# Investigation of Numerosity Representation in Convolution Neural Networks

**Alireza Karami**[1,2]     **Nhut Truong**[2]     **Manuela Piazza**[2]

[1]Cognitive Neuroimaging Unit (UNICOG), INSERM, CEA, CNRS, Université Paris-Saclay, NeuroSpin center
Bâtiment 145, CEA/SAC/DRF/Joliot, Point Courrier 156, F-91191 Gif-sur-Yvette, France

[2]Center for Mind/Brain Sciences (CIMeC), University of Trento
Piazza della Manifattura 1, 38068 Rovereto (TN), Italy

Corresponding author: `alireza.kr@gmail.com`

## Abstract

**Convolutional neural networks (CNNs) have emerged as powerful models for predicting neural activity and behavior in visual tasks. Recent studies suggest that number-detector units—analogous to number neurons—can emerge in CNNs, both in trained networks optimized for object recognition and in untrained networks. In this work, we extend previous studies by investigating whether CNNs encode numerosity at the population level and by examining how the statistical distribution of numerical and non-numerical features in the training dataset influences their internal representations. Recognizing that perceptual systems are finely tuned to the statistical properties of their sensory environment, we compare CNNs trained on both synthetic datasets and a naturalistic dataset that better reflects the real-world conditions shaping human number sense. By systematically manipulating these statistical properties, we assess their impact on the encoding of both numerical and non-numerical features. Finally, we compare these computational representations with those observed in the human brain, highlighting both shared characteristics and key differences that provide deeper insights into the mechanisms underlying numerosity perception in biological and artificial systems.**

**Keywords:** Convolutional Neural Network; Representational Similarity Analysis; Visual Interpretability; Numerosity

**Code:** `https://github.com/alireza-kr/CORNum`

## Introduction

Convolutional neural networks (CNNs) are computational models inspired by early discoveries in biological vision (Fukushima, 1980; Lindsay, 2020). These hierarchical architectures, like the brain, consist of multiple feedforward layers, each comprising artificial units that approximate neuronal processing. Since their introduction, CNNs have emerged as state-of-the-art models for predicting neural activity and behavior in visual tasks (Cichy et al., 2016; Khaligh-Razavi & Kriegeskorte, 2014; Kubilius et al., 2019; Yamins & DiCarlo, 2016; Yamins et al., 2014). Notably, CNNs trained on object classification tasks closely resemble neural responses in the inferior temporal cortex (IT) of both humans and monkeys, a key region for object recognition (Khaligh-Razavi & Kriegeskorte, 2014).

But what happens when images contain multiple objects? The ability to perceive and represent the number of items in a set without counting—known as "number sense"—is widely regarded as a fundamental and evolutionarily ancient cognitive skill shared by humans and many animal species (Dehaene, 2011). This capacity provides significant adaptive advantages for non-human animals (Nieder, 2020). Evidence from human psychophysics (e.g., Burr & Ross 2008), brain imaging in humans (e.g., Piazza et al. 2004; Castaldi et al. 2019; Karami 2024), and single-neuron recordings in animals (e.g., Nieder & Miller 2003; Wagener et al. 2018; Kobylkov et al. 2022) suggests that numerosity is automatically represented in the brain. This may be facilitated by the presence of "number neurons," which are tuned to different numerosities and have been observed in both humans and animals.

Recent findings suggest that number-detector units—analogous to number neurons recorded in the monkey prefrontal and parietal cortex—emerge in the final layer of a CNN trained for visual object recognition (Nasr et al., 2019) and even in an entirely untrained CNN (Kim et al., 2021). Notably, these studies found that these number-selective units were not influenced by non-numeric visual features. Additionally, Zhou et al. (2021) demonstrated that CNNs exhibit numerosity underestimation in connected dot patterns, a phenomenon previously observed in humans (Franconeri et al. 2009; He et al. 2009). These findings align with evidence of specialized neurons that respond to the number of items in visual displays in numerically naive monkeys (Viswanathan & Nieder, 2013), crows (Wagener et al., 2018), untrained 10-day-old domestic chicks (Kobylkov et al., 2022), and even 3-month-old infants (Gennari et al., 2023). Collectively, this evidence suggests that numerical representation arises from intrinsic processes built into the visual system—processes that can also emerge in CNNs.

Much of the debate in the literature on visual numerosity perception concerns whether numerosity perception relies on a dedicated neurocognitive system or emerges from a more general magnitude system that derives number estimates from low-level visual properties such as item area, surface area, total field area, and item density (Leibovich et al., 2016). Even with carefully designed stimuli, fully controlling

all continuous visual variables simultaneously is impossible (DeWind et al., 2015). To address this, recent fMRI (Castaldi et al. 2019; Karami 2024) and MEG (Karami, 2024) studies have employed representational similarity analysis (RSA) at the population level—using the combined activity of many voxels or channels—to disentangle the contributions of numeric and non-numeric features to neural representations. Following this approach, we investigate numerosity at the population level, allowing us to capture the distributed coding of numerical information across neural network models and clarify the complex interplay between numerical and non-numerical features.

Neural network models are often trained on visual stimuli that do not accurately reflect the statistical structure of our developmental environment (Mehrer et al., 2021). Many deep learning models of numerosity perception, for example, rely on synthetic datasets where images are generated pixel-by-pixel with all numerosities occurring at equal frequency (Stoianov & Zorzi 2012; Testolin, Dolfi, et al. 2020; Mistry et al. 2023) or on real images that predominantly feature single objects (Nasr et al., 2019). These datasets typically decouple numerosity from specific non-numerical features, overlooking the complex correlations found in natural scenes. Since perceptual systems are finely tuned to the statistical properties of their sensory environment (Fiser et al., 2010), it is crucial to train models on data that better reflect these conditions. To address this, we trained CNNs on datasets with different statistical distributions—including synthetic datasets where numerical and non-numerical features were systematically controlled, as well as a naturalistic dataset featuring multiple objects derived from real visual scenes—to examine how these variations shape the encoding of numerical and non-numerical features.

## Methods

To assess whether CNN models can capture how the human brain represents numerosity beyond non-numerical features at the population level, we used CORnet-Z, a lightweight model with four anatomically mapped areas (V1, V2, V4, and IT) followed by a decoder layer. CORnet-Z is the simplest network in the CORnet family and serves as an efficient alternative to AlexNet. Each anatomically mapped area consists of a single convolution, followed by a ReLU nonlinearity and max pooling, while the decoder is a 1000-way linear classifier (Kubilius et al., 2019). We selected CORnet-Z because it is the simplest and fastest model in the CORnet family, making it a useful starting point for isolating the minimal architectural requirements for numerosity representation. Unlike its more complex counterparts—CORnet-RT, a recurrent extension of CORnet-Z, and CORnet-S, which incorporates ResNet-like skip connections—CORnet-Z has a shallow, feedforward architecture that still captures key aspects of hierarchical visual processing. By starting with this minimal configuration, we aim to identify the extent to which numerosity representations can emerge purely from structural and architectural constraints, without relying on more complex features such as recurrence

or deep residual pathways. This simplicity allows us to better isolate which aspects of the architecture are necessary and sufficient for numerosity-sensitive representations to emerge. We used five versions of CORnet-Z:

1. The completely untrained version with randomly initialized weights to reveal the effect of architecture alone (Cichy et al., 2016).

2. A version trained on object recognition using the ImageNet dataset (Deng et al., 2009), which consists of 1.2 million images spanning 1,000 object categories (Krizhevsky et al., 2012). This version was included to align with previous work by Nasr et al. (2019).

3. A version trained on a numerosity task, where the model learned to associate an image of dots (the stimulus) with the corresponding numerosity. We generated a dataset of visual stimuli containing between 6 and 29 dots using the method described by DeWind et al. (2015), which systematically controls two orthogonal dimensions—"size" and "spacing"—alongside numerosity (see Figure 1.A for sample images). In this approach, size is defined as a linear combination of the logarithm of total surface area and average item area, while spacing is derived from a linear combination of the logarithm of total field area and the inverse of density. The model was trained using Stochastic Gradient Descent (SGD) with a learning rate of 0.001, and the implementation was carried out using custom PyTorch code, available on the paper's GitHub page.

4. Another version was trained on the same numerosity task and number range but using a different dataset, the Natural dataset (Testolin, Zou, & McClelland, 2020). This dataset originates from computer vision datasets created for the PASCAL detection challenge (Everingham et al., 2009), which include images annotated with rectangular bounding boxes representing object sizes and positions. To generate the stimuli, objects in each image were replaced with their corresponding bounding boxes, displayed as non-overlapping white rectangles on a black 30 × 30 pixel background while maintaining their original spatial arrangement as closely as possible (see Figure 1.A for examples).

5. A version trained on the same numerosity task with the same range of numbers as described above, using the ISA2 dataset (Testolin, Zou, & McClelland, 2020). The ISA2 dataset was designed to overcome a key limitation of the Natural dataset, where images with more than 10 objects were rare, ensuring better representation of higher numerosities during training. Additionally, it examines whether deep networks, like the human visual system, benefit from exposure to irregularly shaped objects rather than just rectangles and squares (see Figure 1.A for sample images). To achieve this, the ISA2 dataset includes ellipsoids with varying aspect ratios. The number distribution follows a power-law pattern, meaning smaller numbers appear more

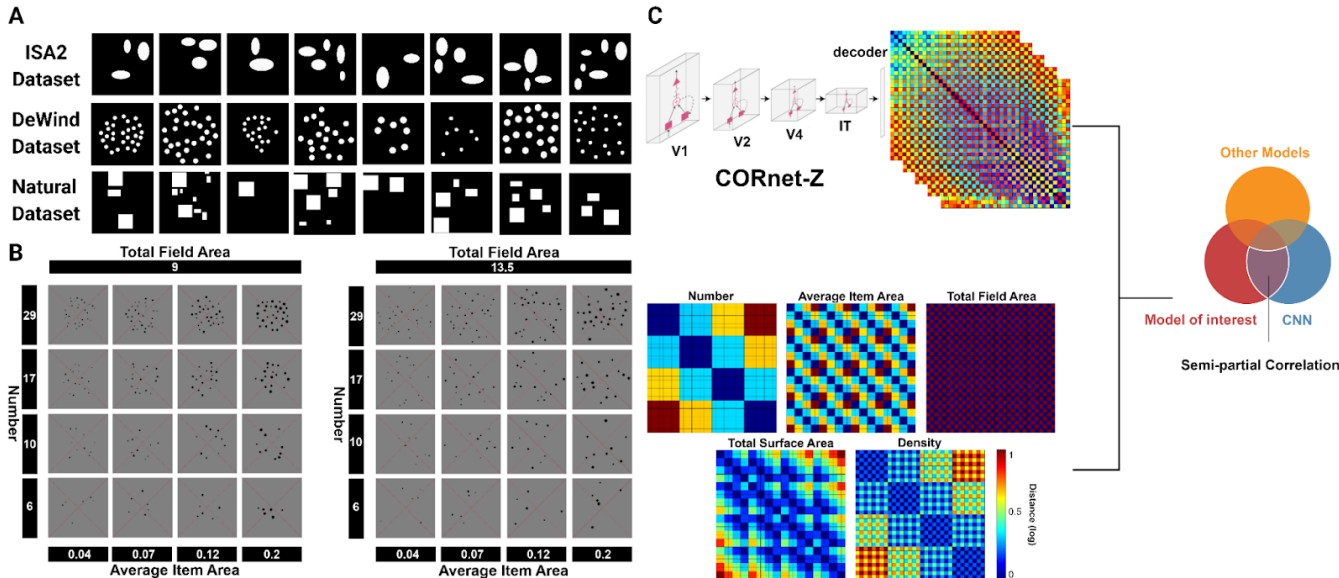

Figure 1: (A) Examples of visual stimuli from the three different datasets used to train CORnet-Z. Each dataset has a distinct statistical distribution of non-numeric features, allowing us to investigate how these features influence the emergence of numerosity representations in the computational model. (B) An illustration of the full set of stimulus conditions. Arrays of dots with four different number and four average item areas displayed within two total field areas. (C) Representational dissimilarity matrices (RDMs) extracted from a given CNN were subjected to a semipartial correlation analysis. In the semipartial correlation analysis, five model RDMs (number, average item area, total field area, total surface area, and density) were used as predictors.

frequently than larger ones—an organization that mirrors how numbers naturally occur in written language (Dehaene, 1992; Piantadosi, 2016).

To test how numeric and non-numeric features are represented in CNNs, we followed the method used in the Karami (2024) fMRI experiment to generate stimuli for testing how numerical and non-numerical features are represented in the human brain. The visual set of dots was orthogonally varied in number, average item area, and total field area. This resulted in 32 conditions, created by crossing four numerosities (6, 10, 17, or 29 dots), four average item areas (0.04, 0.07, 0.12, 0.2 visual square degrees), and two total field areas. The dots were arranged to fit within a small or large total field area, defined by a virtual circle with a diameter of either about 9 or 13.5 visual degrees (Figure 1.B).

All five models were presented with 100 images for each of the 32 conditions. Each image, sized 300 × 300 pixels, served as the input to the network. We selected four layers (V1, V2, V4, and IT) of the network, which are analogous to visual brain areas, and extracted the activation of all nodes in each layer. The activation was extracted using the THINGSvision toolbox (Muttenthaler & Hebart, 2021), and the results from the 100 instances of each condition were averaged to create one activity vector for each condition from each layer's output. We used Pearson correlation to build the CORnet-Z's Representational Dissimilarity Matrices (RDMs).

## Comparing convolutional neural networks with predictor models

To determine whether numerical and non-numerical visual features are represented independently across different network layers, we used an approach based on RSA combined with semipartial correlations. Here's how our method works:

1. Creating Predictive Matrices: We first constructed five RDMs, each corresponding to a different visual feature: number, average item area, total field area, total surface area, and density. Each matrix quantifies the logarithmic difference between every pair of stimuli for a given visual feature: number, average item area, total field area, total surface area, or density.

2. Isolating Unique Contributions: To isolate the contribution of each feature independently of the others, we employed semipartial correlation. This statistical method calculates the relationship between one predictor RDM and the network's own RDM (derived from its activations), while controlling for the shared variance of the other predictors. In simpler terms, it tells us how much a specific visual feature (such as number) uniquely explains the neural activation patterns, without interference from other features like average item area, total field area, total surface area or density.

3. Assessing Statistical Significance: Next, we evaluated whether the observed semipartial correlations were statistically significant. To do this, we performed a permutation

test following the method outlined by Nili et al. (2014). In this test, we randomly shuffled the labels of the stimuli, recalculated the RDMs for each CORnet-Z layer using these permuted labels, and repeated this process 50,000 times. This generated a null distribution of semipartial correlation values representing what would be expected by chance if there were no real relationship between the network's activations and the visual feature predictors. We then compared our actual semipartial correlation values to this null distribution, rejecting the null hypothesis (i.e., concluding that the feature explains a unique part of the variance) if our observed correlation was within the top 5% of the null distribution. This corresponds to using a false-positive rate of 0.05, as described by Kriegeskorte (2008).

## Exploring the latent similarity structure of convolutional neural networks

To examine the latent similarity structure of each network layer's RDM, we applied multidimensional scaling (MDS; Kruskal 1964) using the MATLAB function *cmdscale* and visualized the first two dimensions of the MDS output. This method arranges stimuli in a two-dimensional space, where the distances between them correspond to differences in the response patterns they elicit. Consequently, stimuli positioned closer together in the plot indicate more similar response patterns (Nili et al., 2014).

## Exploring which parts of stimuli influence the network's decision

To see which parts of an image the network uses to make its decision, we used a method called Score-CAM (Wang et al., 2020). This method works by covering up parts of the image and checking how much the network's decision changes. Each feature map, which is a filtered version of the input image that highlights specific patterns, gets a score for how important it is. These scores are then combined to create a heatmap—a colored overlay that shows the most important areas. We did this for different layers of our network (V1, V2, V4, and IT). We chose Score-CAM over another common method called Grad-CAM (Selvaraju et al., 2017) because Grad-CAM usually highlights only one object, while Score-CAM can show several objects at once, which is better for our task of distinguishing numbers.

## Results

### Results of representational similarity analysis on CNN layers

Figure 2 reveals that deeper layers of the network tend to encode numerical information robustly. In the untrained CORnet-Z model, this numerosity signal is significant primarily in the IT layer, suggesting that even without task-specific training, the architecture itself can capture some numerical properties. In contrast, when the network is trained—whether on ImageNet, Natural, ISA2, or DeWind datasets—the contribution of numerosity becomes significant in both the V4 and IT

layers ($p < 0.05$). This indicates that training reinforces and possibly shifts the encoding of numerosity to earlier points in the processing hierarchy.

A similar pattern is observed for total field area. Across all models, the strength of representation for total field area increases progressively up to the V4 layer. However, in the IT layer of the networks trained on the Natural and DeWind datasets, this strength drops significantly. This pattern suggests that while mid-level layers (like V4) integrate information related to spatial extent, the final stages of processing might prioritize other features over total field area—at least in networks exposed to naturalistic or synthetic dot stimuli. In contrast to numerosity and total field area, density shows a different trend. It is more strongly represented in the early layers of the network and then gradually diminishes in deeper layers, regardless of the training condition. Meanwhile, the representation of total surface area remains fairly stable and consistently above zero across all network layers, with a slight enhancement in V1 and IT for most models. This indicates that surface area is a feature that is reliably captured across the processing hierarchy.

Notably, average item area is not prominently represented in any network layer except in the IT layer of the ImageNet and ISA2 models. This observation is in line with human fMRI studies (Castaldi et al., 2019; Karami, 2024), which also report a lack of distinct representation for average item area in most regions of the visual cortex.

Overall, these findings illustrate that the encoding of numerical information—and its separation from non-numerical visual features—is strongly influenced by both network architecture and training. While early layers tend to capture basic visual properties like density, deeper layers—especially when optimized through training—are more specialized in abstract representations such as numerosity, echoing some but not all aspects of human visual processing.

### Results of applying multidimensional scaling on layers of CNNs

Figure 3 shows that when we apply multidimensional scaling (MDS) to the network's representations, the stimuli are generally arranged so that numbers increase along the second dimension in most layers. However, in the IT layer of the networks trained on the Natural and DeWind datasets, this ordering instead appears along the first dimension. This arrangement is reminiscent of the human "mental number line," in which smaller numbers are typically associated with the left side and larger numbers with the right (Galton, 1880; Dehaene et al., 1993). Although this observation is compelling, these results should be interpreted with caution, as the original MDS analysis did not control for potential confounding visual features. We chose this approach to directly compare the network's MDS outcomes with those derived from different brain regions in Karami (2024). We also conducted an additional control analysis to assess whether this observed number line persists when low-level visual features are accounted for (see Supplementary Materials for details). Specifically, we par-

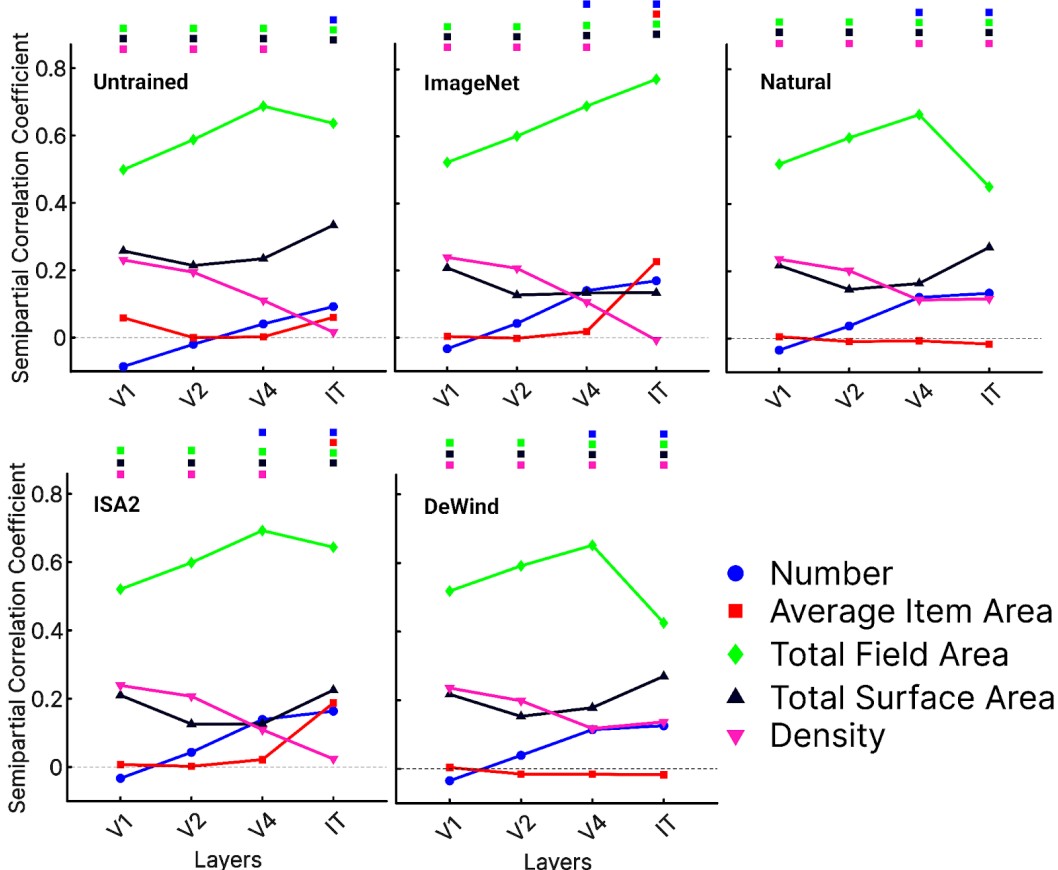

Figure 2: Semipartial correlation coefficients obtained from representational similarity analysis for numerosity, average item area, total field area, total surface area, and density across different layers of untrained CORnet-Z, as well as CORnet-Z models trained on ImageNet, Natural, ISA2, and DeWind datasets. Data points represent the mean semipartial correlation coefficients. Colored markers above the figure indicate where the effect is significantly greater than zero (p < 0.05).

tialled out non-numeric confounds from the RDMs using multiple regression, reconstructed a symmetric residual matrix, and applied MDS to visualize the resulting structure. Notably, a linear spatial organization still emerges in the network's layers in most cases, supporting the existence of a number line that is not fully explained by low-level visual attributes.

Additionally, the MDS plots reveal a clear separation between stimuli with large and small total field areas across all network layers, suggesting that the network distinctly encodes differences in this feature. In contrast, previous studies in higher associative brain regions of the ventral and dorsal streams have reported a much weaker separation based on total field area (Castaldi et al., 2019; Karami, 2024).

**Results of applying Score-CAM on layers of CNNs**

In Figure 4, we observe that from V1 to IT, the highlighted areas progressively expand, shifting from small circles around the dots to broader heatmap patches. This is a natural outcome of feature map sizes shrinking in deeper layers, which leads to deeper-layer units representing larger regions of the image. In all models, V1 and V2 effectively capture all the

dots, suggesting they extract the necessary information for later classification. Interestingly, the untrained model also captures the dots well, but the emphasis is weaker compared to the DeWind model, as seen in the background color of V1. The ImageNet-trained model exhibits a similar pattern to the untrained model. Overall, the main difference among models appears in the IT layer, which is the final input before the classification stage, suggesting that the training method influences number representation mainly in the final layer, rather than in earlier ones.

Specifically, in the IT layer, the model trained on ImageNet highlights individual dots more than the other two models. This matches the observation in Figure 2, where the ImageNet model becomes more sensitive to average item area in IT. This suggests that training on ImageNet leads the network to emphasize discrete object-level features - particularly those associated with average item area. On the other hand, the DeWind-trained and Untrained models show more spread-out heatmaps, focusing on groups or clusters of dots. This wider focus fits with their stronger increase in total surface area in Figure 2 (IT), pointing to a different way of encoding numbers.

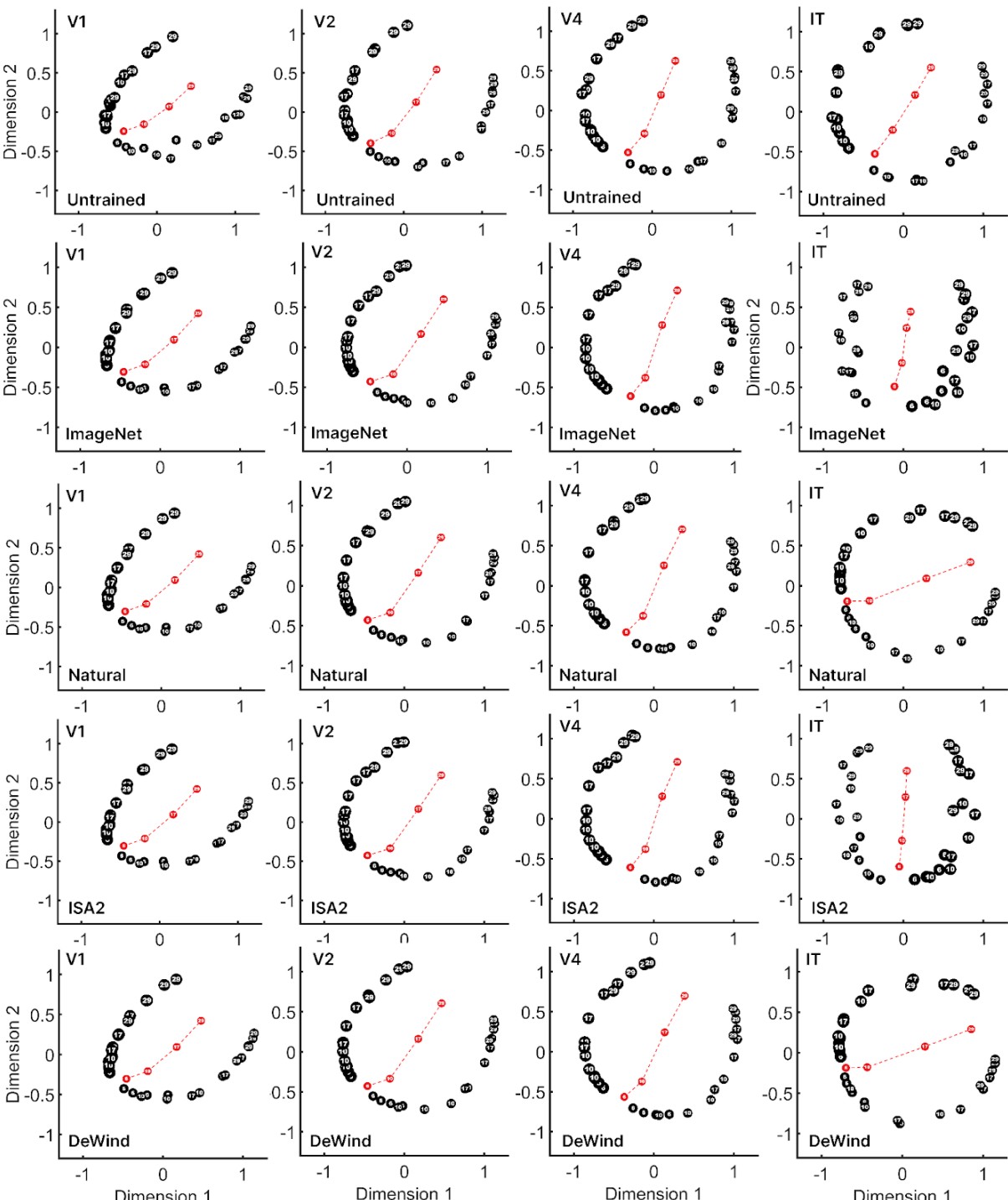

Figure 3: Multidimensional scaling (MDS) reveals representational similarities between stimuli in a two-dimensional space for four layers (V1, V2, V4, and IT) of untrained CORnet-Z, as well as CORnet-Z models trained on ImageNet, Natural, ISA2, and DeWind datasets. The black circles represent the 32 stimuli. The circle sizes vary, indicating stimuli with small total field area (small circles) and larger total field area (large circles). The red circles indicate the average coordinates of each number.

We also quantify the similarities across training regimes (DeWind, ImageNet, Untrained) and layers (V1, V2, V4, IT). Using 100 images (10 first images per 10 numerosities in DeWind dataset), we computed Pearson correlations between flattened (1D) heatmaps for pairs of training regimes. ImageNet-trained heatmaps consistently showed the lowest correlation with the other regimes across layers. This difference was most significant in IT, where average correlations (Mean±Std) were: DeWind vs ImageNet: 0.81±0.17; DeWind vs Untrained: 0.98±0.01; ImageNet vs Untrained: 0.82±0.16. This divergence increased when using only the top 25% heatmap intensities (IT layer): DeWind vs ImageNet: 0.67±0.28; DeWind vs Untrained: 0.95±0.03; ImageNet vs Untrained: 0.68±0.28. For V1-V2-V4, correlations remained very high ($> 0.9$) in all comparisons, in both cases of keeping all pixels and thresholding top 25% pixels. Overall, the quantification results further reveal the similarities and dissimilarities in model mechanisms resulting from different training modes.

## Discussion

In this study, our primary goal was to investigate whether convolutional neural networks (CNNs)—specifically CORnet-Z—encode numerosity at the population level, and how this representation is influenced by different training regimes. We evaluated five variants of CORnet-Z: an untrained model, a model pretrained on ImageNet, and three models trained on numerosity tasks using datasets with varying statistical structures (DeWind, Natural, and ISA2). To assess numerosity representation and its separation from non-numerical visual features, we employed representational similarity analysis (RSA) with semipartial correlations, multidimensional scaling (MDS), and Score-CAM visualizations. Our results showed that numerosity information consistently emerges in deeper layers (especially V4 and IT) across all models, including the untrained one. We also found differences in representational geometry between the networks and human brain data, particularly in how non-numeric features like total field area are encoded.

Many studies have explored how computational models can represent numerosity information extracted from visual images (Dakin et al., 2011; Dehaene & Changeux, 1993; Hannagan et al., 2018; Kluth & Zetzsche, 2016; Knops et al., 2014; Park & Huber, 2022; Paul et al., 2022; Stoianov & Zorzi, 2012; Testolin, Dolfi, et al., 2020; Verguts & Fias, 2004). While one study examined how training a CORnet-S network on a numerosity task with a synthetic dataset reorganizes number-selective units (Mistry et al., 2023), few have investigated how numerical and non-numerical information are simultaneously represented. A notable exception is Testolin, Dolfi, et al. (2020), who used representational similarity analysis and t-SNE to study numerosity in deep belief networks—models that develop internal representations from sensory data (Zorzi et al., 2013). Unlike our approach, which trains CORnet-Z using supervised learning, Testolin, Dolfi, et al. (2020) used

unsupervised learning, thought to be more biologically plausible (Cox & Dean, 2014; Zhuang et al., 2021), as infants naturally learn without labeled data (Bergelson & Swingley, 2012; Frank et al., 2021). However, supervised models more closely match representations in the ventral cortex (Khaligh-Razavi & Kriegeskorte, 2014). Both our study and Testolin, Dolfi, et al. (2020) found that numerosity is represented alongside other visual features. In Testolin, Dolfi, et al. (2020), numerosity became the most dominant feature after training, supporting the idea that humans learn to focus on number while filtering out irrelevant features (Piazza et al., 2018). In contrast, our network emphasized total field area more than numerosity, similar to the early training stages in Testolin, Dolfi, et al. (2020). However, training CORnet-Z on a numerosity task improved its ability to represent numerosity, as reflected in higher semipartial correlation values in the trained vs. untrained networks. This finding aligns with fMRI studies showing sharper numerosity tuning in adults (Piazza et al., 2004) compared to preschoolers (Kersey & Cantlon, 2016).

### How training dataset shapes numeric and non-numeric network representations

In this study, we examine how numerosity information is represented in both an untrained network and a network trained on two types of tasks: object recognition and numerosity. For the numerosity task, we trained the network on both a synthetic dataset—where all numerosities appear with equal frequency, and numerical and non-numerical features are mostly uncorrelated, which does not necessarily reflect natural statistics—and on two datasets intended to capture the statistical properties of real-world environments. Our results show no clear distinction between the representation of numeric and non-numeric features in the network trained on the synthetic DeWind dataset and the Natural dataset. This outcome contrasts with efficient coding models of perception (e.g., Gold & Stocker 2017), which suggest that our limited perceptual capacity is fine-tuned to the common patterns found in natural scenes. In simple terms, these models propose that our brains are optimized to pick up on the regular relationships between numbers and other visual features—like size, spacing, and arrangement—that typically occur in real-world environments. According to this idea, when we look at a group of objects, our brain's sense of number is shaped by the natural patterns it has learned, which help us understand the world more efficiently. Based on this, changing the statistics of the training data should result in different ways of representing numeric and non-numeric features. This is because, in the context of efficient coding models, the model learns from the patterns in the data and focuses on different visual aspects of the dot arrangement. This finding also aligns with human studies, which suggest that visual numerosity perception does not necessarily improve when perceiving real-world scenes compared to artificial displays (Odic & Oppenheimer, 2022).

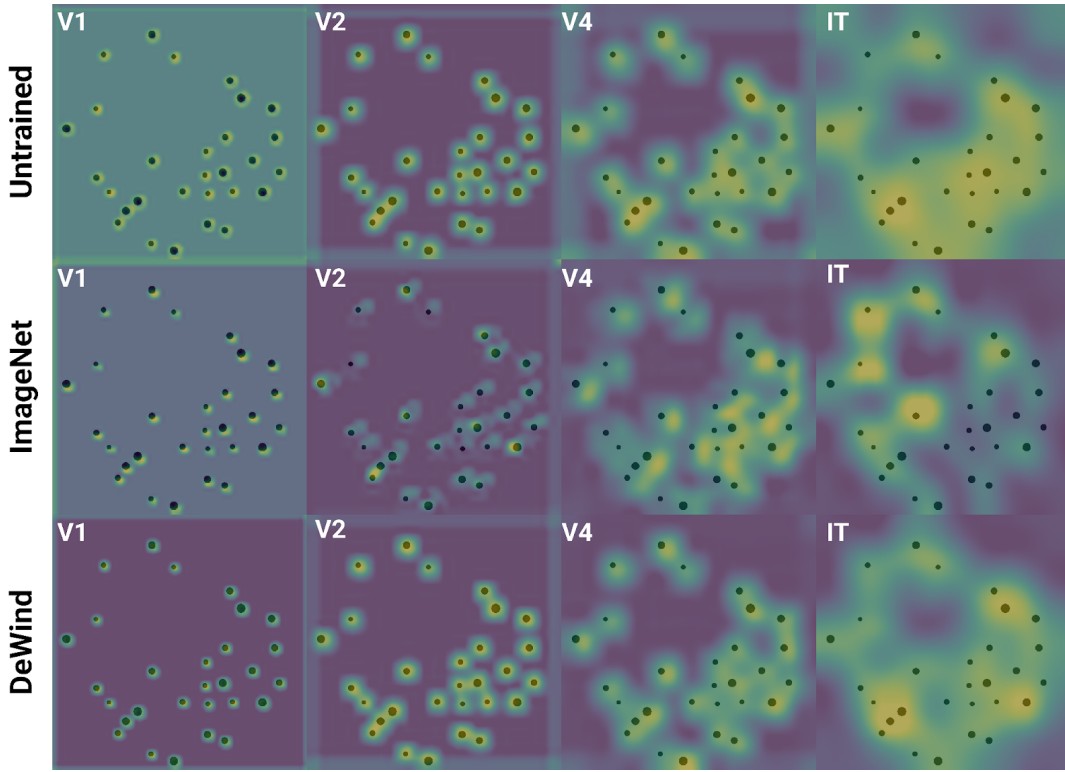

Figure 4: Results from Score-CAM analysis of different layers of CORnet-Z (V1, V2, V4, and IT) on an example image. The visualizations highlight the regions of the image that are important for number classification, with warmer colors indicating higher relevance. As the Natural model resembles the DeWind model, and to conserve space, we visualize only DeWind.

## Human brain and networks encode numeric and non-numeric features using distinct geometrical patterns

Despite the ability of both neural networks and the human brain to represent numeric and non-numeric information, there are differences in the geometry of these representations.

By carefully inspecting the multidimensional scaling (MDS) of different layers in our model alongside data derived from various brain regions (Karami et al., 2023), we found an intriguing discrepancy. Specifically, a curved structure observed in the MDS of the intraparietal sulcus (IPS) region in the human brain—a feature attributed to the presence of a decision variable (Nelli et al., 2023)—was not present in any layer of our network. This discrepancy suggests a fundamental difference in the geometry of representation between our model and higher-order brain regions. The IPS, known to host numerosity representations that are explicitly read out during numerical decision-making (Lasne et al., 2018), appears to operate under different representational principles than those in our network.

Furthermore, our results indicate that the geometry of representations in the deeper layers of the network more closely resembles that of early visual areas rather than higher-order regions. This similarity is evident not only in the spatial patterns of the representations but also in the trends observed in total field area. In our network, the total field area expands in a manner akin to early visual processing regions. For example, Karami et al. 2023 documented an increase in total field area from V1 to V2 in the human visual cortex—a trend that we also observe in our network's earlier layers. However, a notable divergence emerges in higher-order visual regions. While human data reveal a decrease in total field area (possibly reflecting a transition from detailed spatial encoding to more abstract, spatially invariant representations of numerosity, as posited by Viswanathan & Nieder (2020), some network conditions do not fully replicate this reduction. Interestingly, our CNNs show a marked decrease in total field area particularly in networks trained for numerosity discrimination using the Natural and DeWind datasets. This decrease may indicate a convergence toward more spatially invariant representations, echoing the abstraction processes identified in biological systems. Yet, the discrepancies between network and brain data imply that current CNN architectures might be missing specific computational mechanisms needed to facilitate these transformations in the cortex. Recent work by Paul et al. 2022 provides additional insight into these issues. Their study demonstrates that early visual areas represent stimuli using contrast-based image statistics, which correlate strongly with perceived numerosity. They argue that spatial frequency and Fourier power components are crucial for forming downstream numerosity representations and that

converting early contrast-based signals into numerosity-tuned responses in higher cortical areas requires nonlinear interactions among these features coupled with divisive normalization to control for image contrast. Although standard CNNs like AlexNet exhibit contrast-based responses similar to early visual areas, they lack the nonlinear transformations necessary for achieving numerosity tuning in higher layers. This discrepancy suggests that incorporating biologically plausible operations—such as divisive normalization (Carandini & Heeger, 2011)—could lead to more human-like transformations of visual input in these networks.

Overall, our study demonstrates that both neural networks and the human brain encode numerosity alongside other visual features, but with distinct geometrical patterns. In humans, numerosity emerges in early visual areas, as revealed by fMRI (Karami, 2024), and unfolds over time as indicated by MEG (Karami et al., 2023), eventually exhibiting unique structural signatures in higher-level regions. In contrast, while our network models capture some aspects of these representations, they do not replicate these higher-level trends. Taken together, these findings suggest that although current CNNs can partially recapitulate early visual processing, they fall short of replicating the hierarchical transformations required for abstract numerosity encoding. Future research should explore the integration of biologically inspired operations—such as divisive normalization—to enhance the fidelity of model-to-brain mappings, particularly in higher layers.

## Declaration of generative AI and AI-assisted technologies in the writing process

The authors used ChatGPT to assist with rephrasing certain sentences during the preparation of this work. They subsequently reviewed and edited the content as needed and took full responsibility for the final published article.

## Code accessibility

All code used for the analyses and experiments described in this paper is openly available at: `https://github.com/alireza-kr/CORNum`.

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

# Supplementary material

## Multidimensional Scaling After Controlling for Non-Numeric Features

To isolate the unique variance associated with a target dissimilarity matrix, we partialled out the shared variance explained by four control dissimilarity matrices: average item size, total surface area, total field area, and density. Specifically, we vectorized the upper triangular portion (excluding the diagonal) of each matrix and performed a multiple linear regression, using the control matrices as predictors and the target matrix as the dependent variable. The residuals from this regression represent the component of the target matrix that is independent of the control matrices. These residuals were then used to reconstruct a symmetric matrix, which served as the input for further analyses such as multidimensional scaling (see Figure 5)

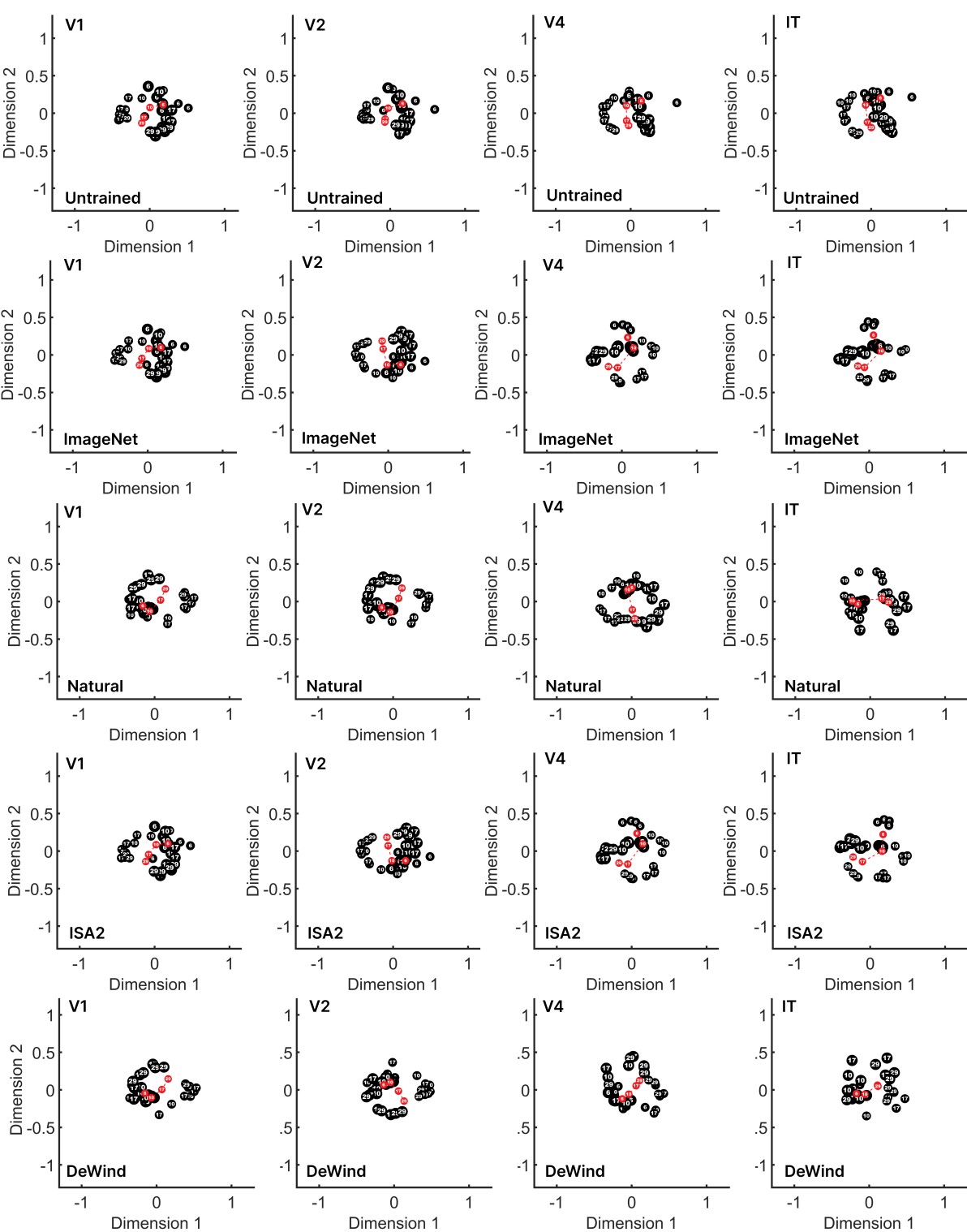

Figure 5: Multidimensional scaling (MDS) reveals representational similarities between stimuli in a two-dimensional space for four layers (V1, V2, V4, and IT) of untrained CORnet-Z, as well as CORnet-Z models trained on ImageNet, Natural, ISA2, and DeWind datasets. The black circles represent the 32 stimuli. The circle sizes vary, indicating stimuli with small total field area (small circles) and larger total field area (large circles). The red circles indicate the average coordinates of each number.

