# OpenReview forum: "Investigation of Numerosity Representation in Convolution Neural Networks"
_ccneuro.org/CCN/2025/Proceedings — CCN 2025 Proceedings asProceedingsPoster_

### Official Review · Reviewer_WFz1 · 2025-03-31
**An interesting approach to studying numerosity representation, but lacking in methodological detail**

**Soundness:** 2
**Clarity:** 2

**Comments:**

[see review comment below for some adjustments]

*Summary:*

The authors studied how well numerosity is represented in the CORnet-Z network. They did this for five different variants of this network (untrained and trained on different dataset types). They then used a variety of analysis approaches to assess how well numerosity is represented in these networks.


*Strengths:*
* The use of a variety of training methods and tasks for the networks considered was well done
* The use of a variety of ways to consider numerosity representation was also done well, and the consistency between the results strengthened the claims
* The topic is of high interest to the CCN community

*Limitations:*
* The analysis methods are poorly explained and I had trouble fully reconstructing how the authors achieved their results (consider for example the currently very brief explanation of the complex diagram in in figure 1C). How their methods work is now described generally in text only, but needs to be a lot more specific and rigorous.
* The results section was too brief for the complexity of the results, and was therefore hard to fully interpret
* Overall, I find that the clarity of the paper needs improvement  --- and as a result can also not fully assess the soundness of the study.

*Questions*
* I was not able to find out how well the different networks performed on their train or test data sets. Was this of any importance for the numerosity results?

*Suggestions*
* I think perhaps to fit in the 8 page page limit it would be a good idea to drop one of the analyses, and give a better explanation of the remaining two analyses .
* I would restructure the discussion section to first provide a summary of the current paper before diving into an in depth comparison with literature, as now
* I found the way the two-column layout was done confusing to read. Specifically, on several pages the two columns on the top of the page follow each other, and the two columns at the bottom of the page. I think it would work a lot better to have the figures placed on top of those pages to avoid this.
* Figure 3 was very hard to parse and understand, I recommend writing a more extensive caption
* page 3 last paragraph: 'Karami et al. (2024) fMRI experiment ' -> 'THE Karami et al. (2024) fMRI experiment'?
* figure 1 caption: 'Representational dissimilarity matrices (RDMs) extracted from CNN were subjected to a' -> 'Representational dissimilarity matrices (RDMs) extracted from A GIVEN CNN were subjected to a'?

**Expertise:**

2

**Interest:**

3

---

> ### Author Rebuttal · Authors · 2025-04-13
>
> ## Complexity of Results and Brevity of Explanation
> Thank you for pointing this out. In the revised version, we have significantly expanded relevant sections of the Methods and Results and provided a clearer walkthrough of each panel in Figures 2 and 3.
> ## Clarity and Explanation of Methods
> The Discussion section now begins with a concise recap of our goals, methods, and key findings before connecting to broader literature.
> Figure captions have been rewritten for clarity. In particular, Figure 3 and 4 now comes with a longer caption explaining what is shown in each figures
> ## Network Performance and Relevance to Numerosity
> While our main focus was on examining the internal representations underlying numerosity in CNNs rather than optimizing or reporting classification performance per se, we did assess the performance of our models on independent test sets in some cases. Specifically, for the models trained on the DeWind and ISA2 datasets, we evaluated them on independent test sets drawn from the same statistical distribution as the training data, and both achieved very high accuracy (approximately 0.99). For the network trained on the Natural dataset, however, we did not have an independent test set dedicated to numerosity discrimination. It remains an open question whether the same performance would be maintained if the models were evaluated on test data with a different distribution from that used in training.
> We acknowledge that while performance is an important metric in many studies, our primary goal here was to investigate the internal representational dynamics—and how they are shaped by training regimes—rather than to benchmark classification accuracy. Similar questions regarding performance and generalization have been addressed in previous work (e.g., [Nasr et al., 2019](doi.org/10.1126/sciadv.aav7903); [Kim et al., 2021](doi.org/10.1126/sciadv.abd6127); [Mistry et al., 2023](doi.org/10.1038/s41467-023-39548-5)). We will make sure this distinction is clearly articulated in the revised manuscript to help contextualize our focus on internal representations over traditional performance metrics.
> ## Formatting, Layout, and Minor Comments
> We have addressed all formatting and minor language issues, including figure placement, caption clarity, and the grammatical suggestions (e.g., “the Karami et al. experiment”). These changes improve both readability and flow throughout the paper.

---

> > ### Comment · Reviewer_WFz1 · 2025-04-17
> >
> > Many thanks for the reply and adjustments. I find both the clarity and soundness quite improved!
> >
> > I would like to clarify that my question about the performance and importance to the result was not just about if the performance was high enough, but I was curious if the representation of numerosity depended in any way on model performance (e.g., perhaps the representations are much stronger if the performance is also high). However, if the performance was consistently so high as 0.99 this seems hard to assess.

---

> > > ### Author Response · Authors · 2025-04-21
> > >
> > > We thank the reviewer for this insightful point. You’re correct that examining whether numerosity representations co‑vary with model performance would provide valuable insight. In our current experiments, overall accuracy was uniformly high (mean = 0.99), leaving essentially no variance on which to perform such an analysis. Consequently, we did not include a performance–representation correlation in this manuscript. We plan to investigate the relationship between task performance and representational strength in future work by introducing conditions with a wider range of difficulty.

---

### Official Review · Reviewer_ePwe · 2025-03-31
**The paper provides an interesting analysis of numerosity representations emerging in a CNN (CORnet-Z) and explores potential parallels to human representations. It effectively demonstrates numerosity encoding in later convolutional layers, consistently across training conditions, and rigorously controls for visual confounders through methods like semi-partial RSA. However, concerns remain regarding confounding influences in MDS analyses and limited interpretability of Score-CAM visualizations. Further empirical comparisons to human neural data (e.g., fMRI or MEG) would significantly strengthen the proposed neural parallels.**

**Soundness:** 1
**Clarity:** 2

**Comments:**

## Summary

The authors investigate numerosity representations within a convolutional neural network (CNN) architecture (CORnet-Z). They train multiple instances of the model: an untrained version, a version pretrained on ImageNet, and versions trained on synthetic numerosity datasets (DeWind, Natural, ISA2). They then evaluate the learned representations on a separate stimulus set designed to control for confounding factors that could influence numerosity effects, such as image density or total surface area. To disentangle these factors, the authors apply representational similarity analysis (RSA) with a semi-partial correlation approach.

The main claims of the paper are:

- CORnet-Z encodes numerosity at the population level (through distributed activations across units) predominantly in later convolutional layers (V4, IT). This numerosity encoding occurs consistently across all datasets, including in an untrained instance.

- A visual number line emerges within the CORnet-Z representations, as demonstrated by multidimensional scaling (MDS), indicating numerically closer counts cluster in representational space.

- Image-level visualizations suggest the network representations capture the precise locations of each number item.

## Strengths

The paper explores a topic in computational cognitive neuroscience that is of interest to a broader audience at the conference, specifically whether numerosity emerges naturally in deep networks, how this representation is influenced by training and what the similarities are to previous findings in humans. The paper was written well, and I could follow it easily. The authors apply a broad range of elaborate methods, ranging from multivariate pattern analysis to sophisticated DNN visualization techniques. I think the semi-partial RSA results elegantly isolate numerosity effects from confounding visual properties. While I had not been aware of semi-partial RSA beforehand, it seems to me that this is a solid methodological choice. Although the effects for numerosity seem small, they are indeed significant, and Figure 2 also shows a numerosity effect that I would expect since it only emerges in later convolutional representations.

## Limitations

### Major comments

I have some comments regarding the other two analyses of the paper, and I would like to see these points addressed to improve my rating.

- Although the authors carefully control for confounding factors in their RSA analysis, it seems to me that similar controls are not applied to the MDS plots. If confounding factors (e.g., density) correlate highly with numerosity (as suggested by the RDM plots in Figure 2c), might it be possible that the apparent number line shown by MDS reflects these confounders rather than numerosity itself? If this is not the case or if I have misunderstood, I would like to see clearer phrasing explaining why MDS can indeed isolate numerosity effects. Otherwise, I am unsure if the MDS visualizations truly support the claims made in the paper.

- The interpretation and utility of the Score-CAM visualizations seem limited to me. Early CNN layers inherently act as edge detectors. Since Score-CAM is gradient-based, the visualizations might primarily reflect foreground-background segmentation rather than genuine numerosity encoding. It would help if the authors clarified the specific added value of this analysis beyond demonstrating typical CNN responses to edges in early layers and larger receptive fields in later layers.

### Minor Comments

- Throughout the paper, the authors make direct comparisons to human numerosity encoding in ventral and dorsal visual streams without incorporating empirical neural data (e.g., from fMRI or MEG). Without such data, it seems that some claims about similarities between numerosity representations in the CNN and visual cortex are somewhat arbitrary and not sufficiently supported by the data.

- The authors limit their analysis to a convolutional architecture (CORnet-Z). It remains uncertain if the numerosity representation observed arises specifically due to the convolutional structure (e.g., local feature sensitivity of kernels) or if numerosity is universally represented in deep neural networks. Including comparisons with non-convolutional architectures (such as transformer-based models) would strengthen the claims, especially the connection to findings in humans.

**Expertise:**

2

**Interest:**

3

---

> ### Author Rebuttal · Authors · 2025-04-13
>
> ## MDS and Potential Confounding Factors
> We used MDS to visually compare representational geometries across CORnet-Z layers and brain regions, closely replicating the method from Karami et al. (2023) by not controlling for non-numeric visual features. This allowed for a direct comparison of MDS structures between our model and human brain data.
> While Karami et al. observed a distinct curved structure in higher visual regions—considered a hallmark of numerosity representation—we did not find a similar pattern in CORnet-Z’s deeper layers. The absence of this curved geometry highlights a key divergence between artificial and biological representations.
> ## Interpretation and Utility of Score-CAM Visualizations
> We employed Score-CAM—a perturbation-based rather than gradient-based visualization method—to examine how training regimes affect internal representations. Unlike Grad-CAM, Score-CAM measures the influence of activation maps on output predictions directly, offering clearer interpretability. Prior work has shown that Score-CAM performs more reliably than Grad-CAM, motivating its use in our study.
> Although deeper layers naturally produce broader Score-CAM heatmaps due to larger receptive fields, we focused on their internal structure. In the IT layer (Figure 4), the ImageNet-pretrained model highlighted individual dots, which aligns with its heightened sensitivity to 'average item area' (Figure 2). In contrast, both DeWind-trained and Untrained models exhibited more diffuse attention over dot clusters, reflecting stronger correlations with 'total surface area'. In V1, all models displayed similar heatmaps, providing a consistency check.
> These qualitative visualizations complement our quantitative findings, showing how different training regimes shape numerosity representations. This methodology aligns with explainable AI approaches (Li et al., 2019; Kataoka et al., 2020; Phillips et al., 2020).
> ## Generality Beyond Convolutional Architectures
> Though transformer models merit future exploration, we focused on CORnet-Z—a lightweight, feedforward model—to examine minimal architectural requirements for numerosity. Its simplicity, lacking recurrence (as in CORnet-RT) or skip connections (as in CORnet-S), allows us to isolate the core architectural factors necessary for numerosity-sensitive representations.
> ## Comparisons to Human Numerosity Encoding
> We will clarify that our findings suggest similarities with human data without implying direct equivalence.

---

> > ### Comment · Reviewer_ePwe · 2025-04-21
> >
> > Thank you for the clarifications. Two issues still prevent me from raising my score.
> >
> > First, the MDS plots are still treated as evidence for a visual number line. Without a control analysis, e.g., correlating the leading MDS dimensions with density or surface‑area RDMs, or rerunning MDS on the semi‑partial RSA residuals, it remains unclear whether the observed structure reflects numerosity rather than low‑level confounds.
> >
> > Second, the Score‑CAM visualizations highlight receptive‑field–like regions, but qualitative heat‑maps alone do not demonstrate number sensitivity. A simple quantitative metric (overlap with dot masks, sparsity, dispersion, etc.) would make the comparison across training regimes convincing. Otherwise, the conclusions drawn from Figure 4 are not yet sufficiently supported.

---

> > > ### Author Response · Authors · 2025-04-21
> > >
> > > Thank you for this thoughtful suggestion. We would like to clarify that our primary motivation for applying MDS to the model RDMs—without initially partialling out non-numeric features—was not to claim definitive evidence for a visual number line, but rather to facilitate a direct comparison with a previous fMRI study (Karami et al., 2024) that used a similar approach to examine the representational geometry in different brain regions. This allowed us to relate our network's internal representations to specific brain regions in a consistent manner. A similar strategy was also adopted by [Thompson et al. (2024)](www.cell.com/neuron/fulltext/S0896-6273(24)00729-3), who compared representational geometries in their network with those derived from the fMRI data reported by Karami et al., (2024), further underscoring the value of this approach for aligning model and brain representations.
> > >
> > > That said, in response to your comment, we have now performed the additional control analysis as requested. Specifically, we partialled out non-numeric confounds from the numerosity RDM using multiple regression, reconstructed a symmetric residual matrix, and applied MDS to visualize the resulting structure. The [results](drive.google.com/file/d/1vVgdJeYr-EMdjb5oyBJ-nQF1x8OJ3muy/view?usp=sharing) show that a linear spatial organization still emerges in the network's layers in most cases, supporting the existence of a number line that is not fully accounted for by low-level visual features.
> > >
> > > Regarding the heatmap analysis, we also managed to add quantification. We analyzed ScoreCAM heatmaps to quantify similarities across training regimes (DeWind, ImageNet, Untrained) and layers (V1, V2, V4, IT). Using 100 images (10 first images per 10 numerosities in DeWind dataset), we computed Pearson correlations between flattened (1D) heatmaps for pairs of training regimes. ImageNet-trained heatmaps consistently showed the lowest correlation with the other regimes across layers. This difference was most significant in IT, where average correlations (Mean±Std) were: DeWind vs ImageNet: 0.81±0.17; DeWind vs Untrained: 0.98±0.01; ImageNet vs Untrained: 0.82±0.16. This divergence increased when using only the top 25% heatmap intensities (IT layer): DeWind vs ImageNet: 0.67±0.28; DeWind vs Untrained: 0.95±0.03; ImageNet vs Untrained: 0.68±0.28. For V1-V2-V4, correlations remained very high (>0.9) in all comparisons, in both cases of keeping all pixels and thresholding top 25% pixels. Overall, ScoreCAM heatmaps (plus quantification) can reveal similarities and dissimilarities in model mechanisms resulting from different training modes.
> > >
> > > We hope these extra analyses improve the soundness of our paper.

---

### Official Review · Reviewer_enwE · 2025-04-01
**This paper investigates how convolutional neural networks (CNNs) encode numerosity at the population level and examines how different training datasets influence these representations.**

**Soundness:** 2
**Clarity:** 3

**Comments:**

Interest: 3
This paper investigates how convolutional neural networks (CNNs) encode numerosity at the population level and examines how different training datasets influence these representations. The work is highly relevant to the CCN community as it bridges computational modeling and cognitive neuroscience, addressing fundamental questions about numerosity perception—a core cognitive ability shared by humans and non-human animals. The comparison between artificial networks and human brain data makes this work particularly valuable for researchers interested in the neural basis of number sense and visual processing.

Soundness: 2
The methodology is sound and well-executed. The authors use CORnet-Z, which has layers that map to visual brain areas, and systematically compare five conditions: untrained network, ImageNet-trained, and three networks trained on numerosity tasks with different datasets (DeWind, Natural, and ISA2). Their analytical approach using representational similarity analysis (RSA) with semipartial correlations effectively disentangles numeric from non-numeric features.

The Score-CAM visualization and multidimensional scaling (MDS) analyses provide complementary evidence that strengthens their conclusions. The comparison with human neuroimaging data is particularly valuable, highlighting both similarities and differences between artificial and biological systems.

The evidence generally supports their main claims that:

Numerosity information is represented in deeper network layers (V4, IT)
The geometry of these representations differs from what is observed in human brains
Training dataset statistics have less influence on numerosity representation than might be expected

Clarity: 3
The paper is well-structured and clearly written. The introduction provides comprehensive background on numerosity perception in humans and animals, as well as previous computational modeling approaches. The methods section details the network architectures, training procedures, and analysis techniques comprehensively.

The figures effectively communicate the key findings, particularly Figure 2 (showing semipartial correlations across layers) and Figure 3 (showing MDS results). The Score-CAM visualizations in Figure 4 provide intuitive insight into how the networks process the stimuli.

One minor suggestion for improvement would be to expand the discussion of why the training dataset statistics did not significantly affect numerosity representation, as this finding contradicts predictions from efficient coding models.

Comments
This paper makes several valuable contributions to our understanding of numerosity perception in artificial systems:

It demonstrates that numerosity is encoded at the population level in CNNs, even in untrained networks, supporting the idea that number sense may emerge from intrinsic properties of visual processing.

The finding that training dataset statistics (synthetic vs. naturalistic) do not significantly affect numerosity representation is surprising and challenges assumptions from efficient coding models.

The MDS analysis reveals an ordered arrangement of numbers along one dimension (similar to a "mental number line" in humans), suggesting some structural similarities with human representations.

The observed differences between network representations and human brain data (e.g., pattern of total field area representation across layers, absence of curved structure in MDS) highlight the limitations of current models.


For future work, I would suggest further investigating why the network doesn't show the same decreasing trend in total field area representation from early to later layers as observed in the human visual cortex. This might provide insights into fundamental differences in how biological and artificial systems extract numerical information.

Overall, this is a well-executed study that makes meaningful contributions to both computational modeling and our understanding of numerosity perception.

**Expertise:**

2

**Interest:**

3

---

> ### Author Rebuttal · Authors · 2025-04-13
>
> ## Dataset Statistics
> Our findings suggest that training dataset have a limited impact on a network’s ability to encode numerosity. This can be explained through two points.
> 1. Architectural Capacity for Numerosity Extraction
> Even untrained networks contain numerosity-related information, implying that the architecture—particularly convolutional and pooling layers—naturally captures features relevant to numerosity. [Park & Huber (2022)](doi.org/10.7554/elife.80990) showed that a simple center-surround network could extract numerosity without training. Thus, when trained on numerosity tasks, networks tend to reinforce representations already supported by architectural biases. This suggests that architecture, more than dataset-specific statistics, drives numerosity encoding.
> 2. Dataset-Dependent Correlational Structures
> While numerosity encoding appears robust, a more detailed analysis is needed to understand how numerosity co-varies with non-numeric features (e.g., size, area). [Hou et al. (2024)](doi.org/10.1007/s00426-024-02064-2) proposed a vision pipeline that segments naturalistic scenes, enabling estimation of numerosity and extraction of object-level metrics. Applying such methods to large datasets allows for exploring how numerosity correlates with other features, and whether these correlations affect network representations.
> ## Total Field Area and Brain Comparisons
> Our networks show patterns in TFA partially consistent with human data. [Karami et al. (2023)](doi.org/10.1101/2023.12.18.571155) observed increasing TFA from V1 to V2, followed by a decrease in higher areas—a trend not always mirrored in our networks. This decrease may reflect abstraction, where localized representations give way to spatially invariant numerosity codes ([Viswanathan & Nieder, 2020](doi.org/10.1162/jocn_a_01548)). In CNNs trained on naturalistic data, we observed TFA decreases in higher layers, suggesting some convergence.
> [Paul et al. (2022)](doi.org/10.1038/s41467-022-29030-z) found that early visual areas encode contrast-based features related to numerosity and that nonlinear transformations and divisive normalization are essential for tuning in higher areas. CNNs like AlexNet resemble early responses but lack these operations. Incorporating biologically inspired mechanisms (e.g., divisive normalization; [Carandini & Heeger, 2011](doi.org/10.1038/nrn3136)) may enhance model–brain alignment in numerosity processing.
>
> _For details, please refer to the discussion._

---

> > ### Comment · Reviewer_enwE · 2025-04-20
> >
> > Thanks for the answer, which confirm my appreciation for this work. It's a very solid paper!

---

> > > ### Author Response · Authors · 2025-04-21
> > >
> > > We sincerely thank the reviewer for their kind words and encouraging feedback. We're glad that the quality and solidity of the work came through, and we appreciate your support.

---

### Meta-Review · Area_Chair_6MoD · 2025-05-06

**Ccn Recommendation:** Accept as Proceedings

**Metareview:**

The high degree of enthusiasm about the breadth and clarity of this paper, and the enthusiasm from two of three reviewers about its soundness, make it a strong paper. The only possible cause for concern is the soundness-related question raised by reviewer ePwe regarding statistical claims resting on MDS. To the best of my ability to tell, the authors have addressed this concern adequately in their final response to this reviewer. I am therefore happy to recommend acceptance of this paper to the CCN proceedings.

**Summary:**

Reviewers consistently praised the interest of this paper with all three rating it “broad” (“highly relevant to the CCN community as it bridges computational modeling and cognitive neuroscience”; “of interest to a broader audience at the conference”; “The topic is of high interest to the CCN community”).

There was also consensus that, after revision, it is very clearly written, with two reviewers rating it “adequate” (“The paper was written well, and I could follow it easily.”; “I find both the clarity and soundness quite improved!”) and one “exceptional” (“The paper is well-structured and clearly written”).

All reviewers found the RSA-based analysis to be technically sound (“The methodology is sound and well-executed”; “it seems to me that this is a solid methodological choice”; “variety of ways to consider numerosity representation was also done well, and the consistency between the results strengthened the claims”).

While reviewers enwE and WFz1 rated the paper as technically “adequate”, reviewer ePwe rated it “inadequate” and raised a technical concern about an analysis involving MDS. The authors engaged in a back-and-forth with this reviewer, which concludes with them implementing a suggested control analysis. Specifically, ePwe suggests: “Without a control analysis, e.g., … or rerunning MDS on the semi‑partial RSA residuals, it remains unclear whether the observed structure reflects numerosity rather than low‑level confounds.”, and authors respond: “we partialled out non-numeric confounds from the numerosity RDM … and applied MDS to visualize the resulting structure. The results … [support] the existence of a number line”. To the best of my understanding, this analysis does adequately address the reviewer’s concern.

**Expertise:**

2